# Curriculum Learning for Biological Sequence Prediction: The Case of De Novo Peptide Sequencing

**Xiang Zhang** [* † 1 2]  **Jiaqi Wei** [* 3 4]  **Zijie Qiu** [* 3 1]  **Sheng Xu** [3 1]
**Nanqing Dong** [3]  **Zhiqiang Gao** [3]  **Siqi Sun** [1 3]

## Abstract

Peptide sequencing—the process of identifying amino acid sequences from mass spectrometry data—is a fundamental task in proteomics. Non-Autoregressive Transformers (NATs) have proven highly effective for this task, outperforming traditional methods. Unlike autoregressive models, which generate tokens sequentially, NATs predict all positions simultaneously, leveraging bidirectional context through unmasked self-attention. However, existing NAT approaches often rely on Connectionist Temporal Classification (CTC) loss, which presents *significant* optimization challenges due to CTC's complexity and increases the risk of training failures. To address these issues, we propose an improved non-autoregressive peptide sequencing model that incorporates a structured protein sequence *curriculum learning* strategy. This approach adjusts protein's learning difficulty based on the model's estimated protein generational capabilities through a sampling process, progressively learning peptide generation *from simple to complex sequences*. Additionally, we introduce a self-refining inference-time module that iteratively enhances predictions using learned NAT token embeddings, improving sequence accuracy at a fine-grained level. **Our curriculum learning strategy reduces NAT training failures frequency by more than 90 %** based on sampled training over various data distributions. Evaluations on nine benchmark species demonstrate that our approach **outperforms all previous methods across multiple metrics and species**. Model and source code are available at Github.

---

[*]Equal contribution [†]work done while interning at Fudan University. [1]Fudan University [2]University of British Columbia [3]Shanghai Artificial Intelligence Laboratory [4]Zhejiang University. Correspondence to: Xiang Zhang <xzhang23@ualberta.ca>, Siqi Sun <siqisun@fudan.edu.cn>.

*Proceedings of the 42nd International Conference on Machine Learning*, Vancouver, Canada. PMLR 267, 2025. Copyright 2025 by the author(s).

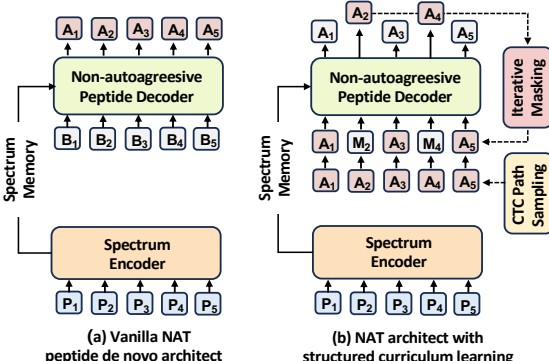

*Figure 1.* Comparison between (a) Vanilla NAT and (b) our proposed NAT peptide sequencing architectures, which integrate curriculum learning and a self-refining module.

## 1. Introduction

Peptide sequencing via tandem mass spectrometry plays a pivotal role in proteomics research, with significant implications for fundamental and applied studies in chemistry, biology, medicine, and pharmacology (Aebersold & Mann, 2003; Ng et al., 2023). The principal aim of peptide sequencing, as illustrated in Figure 2, is to deduce the amino acid sequences of segmented short protein sequences from mass spectra derived from specific biological samples. Database-searching-based sequencing and de novo peptide sequencing are the two most widely used methods for identifying peptide sequences (Chen et al., 2020). Traditional database search methods have inherent limitations, such as the inability to identify peptide sequences absent from the database and the dramatic increase in computational costs and processing times as the database expands. In contrast, de (from) novo (scratch) peptide sequencing directly deduces peptide sequences from the spectra, overcoming the limitations of the static search space in database-based algorithms (Muth et al., 2018).

Deep learning-based models (LeCun et al., 2015; Liu et al., 2022; Zhang et al., 2024) have revolutionized the scientific field, including de novo sequencing. Notably, non-autoregressive Transformer (Gu et al., 2017; Xiao et al., 2023) models have demonstrated superior performance

among all deep learning-based methods in protein sequence predictions (Lin et al., 2023; Hayes et al., 2024; Zhang et al., 2025). Unlike autoregressive models, which rely on "next token prediction" for generation, non-autoregressive models compute the token probabilities at each position simultaneously. This parallel prediction approach enables bidirectional information flow during generation through a self-attention mechanism. Specifically, each position accesses information from all surrounding positions when generating its token, as opposed to relying solely on the preceding ones in an autoregressive model. Such an approach closely aligns with the nature of protein formation and has proven to be more accurate and efficient in de novo sequencing as well as some of the other bio-sequence-related tasks (Eloff et al., 2023; Zhang et al., 2025).

Given that straightforward token-by-token optimization with cross-entropy loss in Non-Autoregressive (NAT) modeling often leads to poor global sequence-level connectivity (Gu et al., 2017) (explained in later section), many NAT-based models employ alternative loss objectives such as Connectionist Temporal Classification (CTC) loss (Graves et al., 2006; Graves & Jaitly, 2014) and Directed Acyclic Transformer (DAT) loss (Huang et al., 2022b). The previous NAT-based de novo sequencing model (Figure 1a) also utilized the CTC objective and has achieved state-of-the-art performance (by 10% improvement compared to AT models). However, the naive CTC objective is far from flawless. It involves complex reduction rules (detailed in later section) and creates a vast search space during optimization, leading to an *unstable learning curve* and *slow convergence* during training (Figure 4). This optimization challenge adversely affects overall performance.

In this work, we enhance standard CTC training and introduce the first protein curriculum learning model for improved sequence prediction. Our method exposes the NAT model to proteins of varying difficulty levels, guiding it to progressively learn from simple to complex sequences, mirroring the human learning process by starting with easier prediction targets and gradually increasing difficulty.

Specifically, we integrate *curriculum learning* with a CTC-based training objective, sampling from the model's own CTC output path during training to determine learning targets and adjust difficulty based on its *current performance* (Figure 1b). By exposing the model to simpler targets first, the curriculum learning approach enables a more gradual and smooth learning process, eliminating the need for the model to grapple with complex protein generation rules from the very beginning.

Furthermore, iterative refinement has proven highly effective in various protein modeling tasks, including protein language modeling and structure prediction (Jumper et al., 2021; Abramson et al., 2024). By incorporating previously

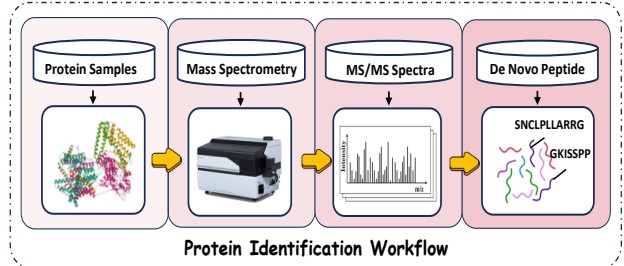

*Figure 2.* Overview of the protein identification workflow. Protein samples are digested into peptides, which are then analyzed using mass spectrometry to generate MS/MS spectra. These spectra are subsequently used in de novo peptide sequencing to determine the peptide sequences.

predicted sequences as input, models can iteratively refine their predictions, improving generation quality. However, traditional NAT models (Zhang et al., 2025) struggle with this approach, as they are trained without conditioning on input sequences beyond positional encoding. Our methodology overcomes this limitation by incorporating input sequence information into the curriculum learning process. By leveraging learned token embeddings, we seamlessly integrate iterative refinement during inference, significantly enhancing the accuracy of protein sequence generation.

Experiments and case studies demonstrate that our training framework effectively mitigates three common failure modes in NAT protein models: loss explosion, extreme overfitting, and unstable loss convergence. Our approach ensures a consistently smooth training curve across diverse datasets, including both well-distributed and poorly distributed samples, achieving over 90% reduction in training failures.

Experiments conducted on the widely recognized 9-species-V1 (Tran et al., 2017) and 9-species-V2 (Yilmaz et al., 2024) benchmark datasets demonstrate the superiority of the proposed RefineNovo over all previous models across various evaluation metrics, establishing a new state-of-the-art in the field. Additionally, we investigate RefineNovo's capacity to differentiate between amino acids with similar masses, showcasing its adeptness in discerning subtle distinctions among challenging amino acids. The overall performance again highlights the exceptional capabilities of RefineNovo, positioning it as a promising and innovative tool in the field of proteomics.

## 2. Related Work

**Autoregressive v.s. Non-Autoregressive.** Autoregressive generation with Transformer models, where tokens are predicted sequentially, has demonstrated outstanding performance across various scenarios (Xiao et al., 2023). However, this approach can be time-consuming, especially for

long sequences. To address this limitation, Gu et al. (2017) introduced the first NAT model for neural machine translation, which facilitates parallel decoding and significantly accelerates the generation process. This substantial increase in inference speed has attracted much attention to NAT methods, leading to impressive progress through techniques such as knowledge distillation (Zhou & Keung, 2020; Ding et al., 2021b; Shao et al., 2022), innovative learning strategies (Qian et al., 2020; Ding et al., 2021a; Zhu et al., 2022), iterative methods (Stern et al., 2019; Guo et al., 2020; Savinov et al., 2021; Huang et al., 2022c), latent variable-based techniques (Ma et al., 2019; Shu et al., 2020; Song et al., 2021; Bao et al., 2021), enhancement techniques (Wang et al., 2019; Yin et al., 2023; Guo et al., 2019; Ding et al., 2020; Liu et al., 2022; Huang et al., 2022a), and different criterion (Saharia et al., 2020; Shao et al., 2020; Ghazvininejad et al., 2020), all of which have further improved the performance of NAT models.

Building on these advancements, NAT models have also shown substantial benefits in biology-related sequence generation tasks, such as protein generative modeling (Lin et al., 2023; Hayes et al., 2024) and de novo peptide sequencing (Zhang et al., 2025), demonstrating the powerful generative ability brought by the bi-directional scheme.

**Deep Learning-Based De Novo Peptide Sequencing.** With the advent of deep learning (LeCun et al., 2015; Gao et al., 2023; Jin et al., 2023), the performance of de novo peptide sequencing has markedly improved. A notable pioneer in this domain is DeepNovo (Tran et al., 2017), the first deep learning-based method leveraging CNNs and LSTMs to model spectra and peptide sequences. Following this innovation, numerous deep learning-based approaches have emerged for de novo peptide sequencing (Zhou et al., 2017; Karunratanakul et al., 2019; Yang et al., 2019; Liu et al., 2023; Mao et al., 2023). However, these methods often involve complex modeling techniques, including the combination of multiple neural networks and intricate post-processing steps (Yilmaz et al., 2022). Casanovo (Yilmaz et al., 2022; 2024) and its derivatives (e.g., AdaNovo (Xia et al., 2024a), HelixNovo (Yang et al., 2024), InstaNovo (Eloff et al., 2023), SearchNovo (Xia et al., 2024c)), and RankNovo (Qiu et al., 2025) introduced the Transformer architecture to directly model the de novo peptide sequencing problem analogously to machine translation in natural language processing. Among these, ContraNovo (Jin et al., 2024) implemented a contrastive learning strategy and integrated additional amino acid mass information, thereby enhancing sequencing accuracy. Recently, PrimeNovo (Zhang et al., 2025) presented the first non-autoregressive Transformer architecture, achieving state-of-the-art results in this task.

PrimeNovo highlights the potential of non-autoregressive decoding in de novo peptide sequencing. In this study, we aim to build on this potential by refining the NAT-based method to further enhance de novo sequencing performance. Our focus is on increasing the reliability of the decoding process, which we believe will lead to advancements in the field.

## 3. Method

### 3.1. Problem Formulation

We first formally define the task of de novo peptide sequencing. Our goal is to translate a given mass spectrometry spectrum into the sequence of amino acids it encodes (Figure 2). The spectrum consists of a set of paired mass-to-charge ratio values (x-axis) with corresponding peak intensity values (y-axis), denoted as $\mathcal{I} = \{(\mathrm{mz}^{(1)}, \mathrm{p}^{(1)}), (\mathrm{mz}^{(2)}, \mathrm{p}^{(2)}), \ldots, (\mathrm{mz}^{(k)}, \mathrm{p}^{(k)})\}$. Additionally, two more pieces of information about the target peptide are provided by the mass spectrometer: the overall peptide mass (precursor mass) $m$ and the entire peptide charge $z$. Using all these inputs, we aim to predict the correct amino acid sequence $\mathcal{A} = (a_1, a_2, \ldots, a_n)$.

### 3.2. Non-autoregressive Transformer BackBone

Our model utilizes an encoder-decoder transformer architecture, similar to the previous NAT-based de novo model (Zhang et al., 2025). To incorporate the curriculum learning strategy, we make several architectural changes, outlined in detail below.

**Spectrum Encoder.** The encoder compresses the spectrum input $\mathcal{I}$ into a meaningful embedding $\mathbf{E}$ in latent space. We treat the values in $\mathcal{I}$ as a sequence and encode each float value of $\mathrm{mz}^{(i)}$ using a sinusoidal encoding as follows::

$$\mathrm{e}_i^0(\mathrm{mz}) = \begin{cases} \sin((\mathrm{mz})/(\frac{(\mathrm{mz})_{\max}}{(\mathrm{mz})_{\min}}(\frac{(\mathrm{mz})_{\min}}{2\pi})^{\frac{2i}{d}})), & \text{for } i \leq \frac{d}{2} \\ \cos((\mathrm{mz})/(\frac{(\mathrm{mz})_{\max}}{(\mathrm{mz})_{\min}}(\frac{(\mathrm{mz})_{\min}}{2\pi})^{\frac{2i}{d}})), & \text{otherwise} \end{cases} \quad (1)$$

where $d$ is the hidden dimension of our model. Similarly, the peak intensity $\mathrm{p}^{(i)}$ is encoded with the same function into $d$ dimensions and then added to $\mathrm{e}(\mathrm{mz})$ at each position. The encoded spectrum embedding $\mathbf{E}^0 = (\mathrm{e}_1^0, \mathrm{e}_2^0, \cdots, \mathrm{e}_k^0)$ is then fed into the Transformer Encoder with $m$ layers, where the $j$th encoder layer updates the last layer embedding $\mathbf{E}^{j-1}$ as follows:

$$\mathbf{E}^j = \mathrm{SelfAttention}(\mathrm{e}_0^{j-1}, \mathrm{e}_1^{j-1}, \cdots, \mathrm{e}_k^{j-1}) \quad (2)$$

The output from the last layer, $\mathbf{E}^{(m)}$, is used as the feature representation of the input spectrum and will be used in the decoding process.

**Peptide Decoder.** Unlike autoregressive decoders, which use self-attention for next token prediction, NAT decoders

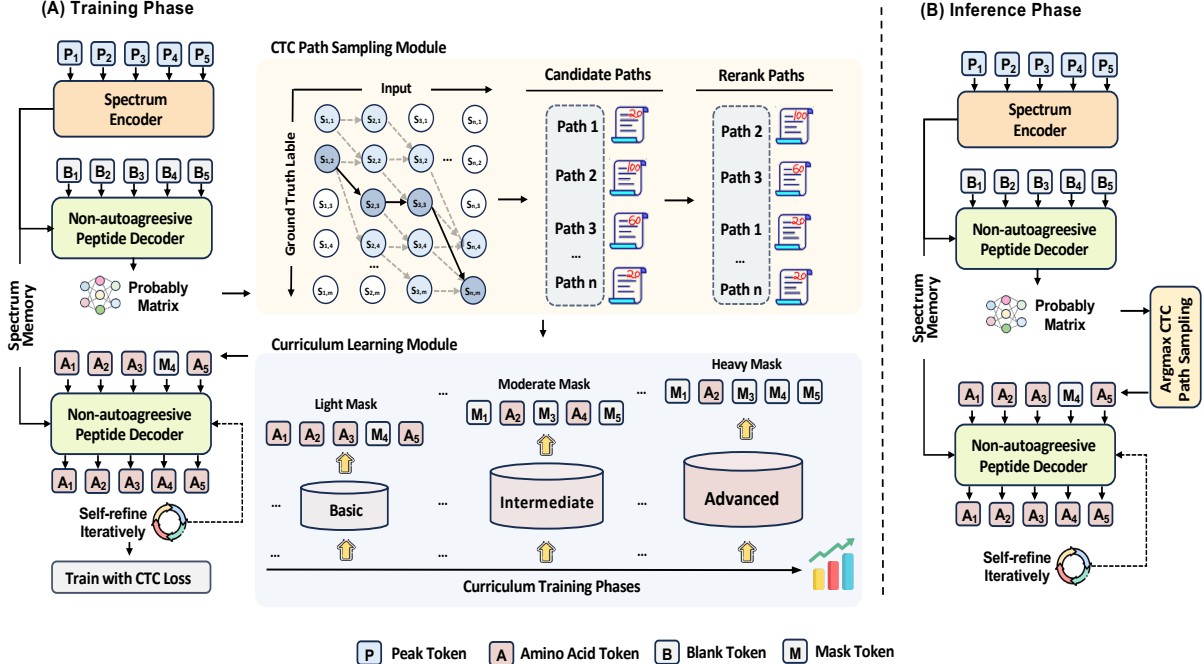

*Figure 3.* The architecture of RefineNovo. (A) The training phase of RefineNovo begins with a Spectrum Encoder that processes the input spectra. Following encoding, the Non-Autoregressive Peptide Decoder leverages the encoded spectra and blank tokens to generate a probability matrix. To predict potential sequences, CTC path sampling identifies all candidate paths, which are then re-ranked for optimal selection. The model employs a Masking strategy, progressively masking tokens to support a curriculum training approach that advances from basic to advanced stages. Further refining the learning process, the model iteratively adjusts based on previously predicted sequences and is trained using the CTC loss function. (B) In the inference phase, the model operates similarly to the training phase, with the primary difference being the use of Argmax CTC path sampling. To maintain accuracy, the model continues to iteratively refine its predictions, ensuring precise peptide sequence generation.

generate token probabilities for each position independently, using a self-attention-based decoder module. Unlike previous NAT designs that only take positional encodings as input, our model incorporates calculated sequences as input for curriculum learning purpose. To accommodate this, we add an embedding layer, denoted as $\mathbf{h}_i^0 = \text{EmbeddingLayer}(y_i)$, where $y_i$ is the input token at the $i$th position. The encoded input is then passed through both self-attention and cross-attention layers with the spectrum features $\mathbf{E}^m$. Finally, the output of the last layer, $\mathbf{h}^{(L)}$, is mapped to the probability of tokens as: $P_s(\cdot \mid \mathcal{I}) = \text{softmax}(W\mathbf{h}_s^{(L)})$ for decoding position $s$.

**CTC Training.** Using a naive cross-entropy loss in parallel prediction models can lead to the multi-modal problem[1] due to the lack of token dependencies (Gu et al., 2017). To address this, we use the Connectionist Temporal Classification (CTC) loss as our optimization objective. In CTC, we first define a maximum generational length $T$, which can be reduced to the target length using the following rules $\Gamma(\cdot)$:

---

[1]For example, translation of "au revoir" to english might result in sentence such as "good you" or "see bye".

1) consecutive identical tokens are merged; 2) placeholder token $\epsilon$ is removed; 3) identical tokens adjacent to $\epsilon$ are not merged. For example, the sequence AABC$\epsilon$C is reduced to ABCC.

During training, instead of maximizing the generation probability of the true token $a_i$ at position $i$, CTC maximizes the probability of all generation paths $\mathbf{y} = (y_1, y_2, \ldots, y_T)$ such that $\Gamma(\mathbf{y}) = \mathcal{A}$. For example, with the target sequence ATC and a generational length of 5, decoding paths such as AATTC and AA$\epsilon$TC will be assigned higher probabilities as they can be reduced to the true sequence. Conversely, the path A$\epsilon$ATTC will be discouraged since it does not reduce to the target sequence.

Therefore, the objective for CTC is to maximize the total probability $P(A|\mathcal{I})$ of all valid paths (those that map to the target sequence $A$), or equivalently, to minimize the negative logarithm of this probability:

$$\mathcal{L}_{CTC} = -\log P(A|\mathcal{I}) = -\log\left(\sum_{\mathbf{y}:\Gamma(\mathbf{y})=A} P(\mathbf{y}|\mathcal{I})\right)$$

The term $P(\mathbf{y}|\mathcal{I})$ represents the probability of a single align-

ment path $\mathbf{y}$. If this path probability is computed as a product of individual token probabilities (e.g., $P(\mathbf{y}|\mathcal{I}) = \prod_{y_i \in \mathbf{y}} P(y_i|\mathcal{I})$ for tokens $y_i$ in path $\mathbf{y}$), then its logarithm, $\log P(\mathbf{y}|\mathcal{I})$, is indeed equal to the sum of the individual log token probabilities, $\sum_{y_i \in \mathbf{y}} \log P(y_i|\mathcal{I})$. This is the property that 'the log of the product of all token probabilities equals the sum of the log of each token's probability.' It's important to note that this property applies to the calculation of the log-probability of a *single path*, whereas the CTC loss involves summing the actual probabilities $P(\mathbf{y}|\mathcal{I})$ of *all* valid paths *before* taking the logarithm. The detailed explanation of CTC loss calculation is in Appendix Section C.

**PMC Unit.** Following the work of Zhang et al. (Zhang et al., 2025), we apply a precise mass control post-decoding unit. Since the precursor mass $m$ provided as input is a strict constraint for the generated sequence $\mathbf{y}$, we use a dynamic programming solver to find the path that satisfies this constraint. Specifically, this is modeled as a knapsack problem where the total generation mass $m$ is the maximum capacity of the knapsack. We select items (amino acid tokens) from each position to fill the bag, with each token having a value (predicted log probability) and a weight (molecular mass). The most valuable (probable) path that meets the mass constraint is determined through a 2D dynamic programming table. The detailed decoding algorithm can be referred to in Appendix Section D.

### 3.3. Protein Curriculum Learning with CTC-based NAT

Parallel generative models face more complex optimization landscapes than autoregressive models due to their larger search space. In autoregressive models, predictions at time step t $t$ are conditioned on previous tokens, i.e.,$P(x_t|x_{1:t-1}, \mathcal{I})$, effectively constraining the search space based on prior information $x_{1:t-1}$ and simplifying optimization. In contrast, NAT models predict $P(x_t)$ independently, without conditioning on previous outputs, leading to a significantly larger search space that makes convergence to the global optimum more challenging. Additionally, the implicit reduction rule in CTC further complicates learning, making target path discovery more difficult than next-token prediction in autoregressive models. As a result, we frequently observe instability in loss convergence and high training failure rates in NAT models (Figure 4).

To address these challenges, we introduce a curriculum learning strategy tailored for CTC-based protein prediction. Unlike naive NAT models, which learn to predict the entire sequence from scratch, our model selectively masks parts of the target sequence$\mathcal{A}$ with a special masking token. This modifies the learning objective from independent sequence prediction to conditioned probability estimation:

$$\mathcal{L} = P(\mathcal{A}|\rho(\mathcal{A}, \mathbf{y}), \mathcal{I}) \qquad (3)$$

where $\rho(\mathcal{A}, \mathbf{y})$ represents the selected unmasked tokens. By "leaking" some ground-truth tokens, prediction becomes easier, and the search space is effectively reduced as:

$$\text{SPACE}(NAT) \sim \gamma \cdot \text{TYPE}(Loss) \cdot \rho_{\text{ratio}} \qquad (4)$$

In this case, $\text{SPACE}(NAT)$ reaches zero when no masking is applied (i.e. $\rho_{\text{ratio}} = 0$, the true label is fully provided) and is maximized when all positions are masked (i.e. Naive NAT model, $\rho_{\text{ratio}} = 1$). The TYPE(ctc) yield much larger search space than TYPE(cross entropy) since each target sequence requires searching through $O(nT)$ pre-ctc-reduction paths, rather than a single path as in cross-entropy loss.

However, while this strategy is straightforward for non-CTC-based learning objectives, it becomes significantly more complex when CTC reduction rules are involved. In CTC, position $t$ in the generated sequence does not necessarily correspond to $a_t$ in the true label as the CTC path is often much longer before reduction. This makes it impractical to create our conditional input $\rho(\mathcal{A}, \mathbf{y})$. To address this, we propose a CTC-based curriculum strategy.

First, we perform a forward pass with an **empty** decoder input to obtain the probability distribution for each position $t \in (1, \cdots, T)$. We then calculate the sequence probability for all valid CTC decoding paths $\mathbf{y}$ such that $\Gamma(\mathbf{y}) = \mathcal{A}$. We re-rank all such paths $\mathbf{y}$ according to their total sequence probability and choose the most likely path $\mathbf{y}'$ as our to-be-masked input $\mathbf{y}'$ (Figure 3 Yellow). We apply masking to obtain the input sequence $\rho(\mathcal{A}, \mathbf{y}')$ which is fed into the decoder for conditioned generation and backpropagation update.

This approach exposes the model to partial information from one valid decoding path, allowing it to infer the structure of the true CTC path while learning to predict masked tokens and generalize to unobserved paths. Intuitively, this partial sequence exposure lowers learning difficulty by strategically "leaking" information, guiding the model to internalize the alignment patterns inherent in CTC decoding.

**Difficulty Annealing.** Naively applying masking (e.g. $\rho_{\text{ratio}} = 0.5$) to reduce learning difficulty can result in ineffective learning due to excessive information leakage. To address this, we adopt a difficulty annealing approach to adaptively reduce the masking ratio over the course of training based on model's per-epoch performance. Specifically, during the forward pass in training time, we calculate the model's prediction accuracy $\text{acc}(\mathcal{A}, \mathbf{y}^{\text{argmax}})$ based on CTC-argmax decoding. The masking ratio is then calculated as $\rho_{\text{ratio}} = \alpha(1 - \text{acc}(\mathcal{A}, \mathbf{y}^{\text{argmax}}))$. Consequently, **most tokens will be unmasked at the beginning of training, with more tokens being masked as the model's predictions become more accurate over time**.

We present the curriculum learning method in Algorithm 1.

---

**Algorithm 1** Curriculum Learning with CTC-based NAT

---

**Require:** Training batch **batch**, decoder **decoder**, peek factor $\alpha$

**Ensure:** Masked oracle sequence **glat_prev**

1: **if** Training mode **then**
2:   # Forward pass without gradient updates
3:   **word_ins_out**, **tgt_tokens** $\leftarrow$ forward_step(**batch**)
4:   **pred_tokens** $\leftarrow \arg\max(\textbf{word\_ins\_out})$
5:   # Compute best alignment paths via CTC
6:   **best_aligns** $\leftarrow$ CTC_align(**word_ins_out**, **tgt_tokens**)
7:   **oracle_pos** $\leftarrow$ midpoint(**best_aligns**)
8:   **oracle** $\leftarrow$ gather(**tgt_tokens**, **oracle_pos**)
9:   **oracle_masked** $\leftarrow$ mask(**oracle**, blank where needed)
10:   # Compute model accuracy via CTC-argmax decoding
11:   **same_num** $\leftarrow \sum(\textbf{pred\_tokens} == \textbf{oracle\_masked})$
12:   **seq_lens** $\leftarrow |\textbf{pred\_tokens}|$
13:   **acc** $\leftarrow 1 - \frac{\textbf{same\_num}}{\textbf{seq\_lens}}$
14:   # Compute adaptive masking probability based on performance
15:   $\rho_{\text{ratio}} \leftarrow \alpha(1 - \textbf{acc})$
16:   # Apply curriculum masking
17:   **mask** $\leftarrow$ random() $< \rho_{\text{ratio}}$
18:   **glat_prev** $\leftarrow$ mask(**oracle_masked**, **mask**)
19: **end if**
  Return **glat_prev**

---

This algorithm outlines the key steps for computing alignment paths, selecting oracle tokens, and applying adaptive masking for training.

The python implementation as well as pseudo-code of this learning can be seen in Appendix Sec. A and Algorithm 1.

### 3.4. CTC-Curriculum-based Protein Iterative Refinement

Our trained model includes a curriculum token embedding layer designed to encode conditional inputs, $\rho(\mathcal{A}, \mathbf{y}')$. However, during inference, this layer is largely unused since there is no ground-truth label, and all conditional inputs are mask-tokens. To maximize its utility, we adopt a multi-pass forward approach to iteratively refine the generated sequence. Notably, prior studies have demonstrated that iterative refinement enhances prediction accuracy in protein-related tasks.

To this end, we propose a whole-sequence refinement inference scheme using the previously trained CTC embedding layer, which can encode meaningful information from any

CTC path $\mathbf{y}$. Specifically, at the $i$-th iteration, the previously decoded pseudo path using argmax, $\mathbf{y}^{(i-1)}$, is used as input for the decoder again for the conditional generation:

$$\mathbf{y}^{(i)} = \arg\max \ P(\cdot|\mathcal{I}, \text{EmbeddingLayer}(\mathbf{y}^{(i-1)})) \quad (5)$$

This process is iterated for $N$ times, with the final output $P(\cdot|\mathcal{I}, \mathbf{y}^{(N-1)})$ sent to the PMC unit for final decoding. We perform whole-sequence recycling instead of decoding partial tokens in each pass because the PMC unit requires a probability distribution for all positions from a single pass to avoid distributional shifts. The algorithm detail is in Appendix Section B.

## 4. Experiments

### 4.1. Datasets

To conduct our study, we utilized three distinct datasets in alignment with previous research for fair comparisons: MassIVE-KB (Wang et al., 2018), 9-species-V1 (Tran et al., 2017), and 9-species-V2 (Yilmaz et al., 2024). MassIVE-KB is a comprehensive collection of human proteomic data, including a high-quality subset of more than 30 million peptide spectrum matches (PSMs). We then evaluated our model on the 9-species benchmark test set, which has been used in all previous de novo sequencing work and serves as a comprehensive evaluation dataset with diverse spectrum-peptide data distributions across nine species. The revised version, 9-species-V2, includes more data samples for each species and enforces a stricter annotation process for higher data quality.

**Evaluation Metrics.** To evaluate our model's prediction accuracy, we used metrics at both amino acid and peptide levels as established by previous research. At the amino acid level, we calculated the count of correctly predicted amino acids, $N^a_{\text{match}}$, using criteria based on mass deviations (Yilmaz et al., 2022). Amino acid Precision was determined by $N^a_{\text{match}}/N^a_{\text{pred}}$. At the peptide level, we considered a peptide accurately predicted if all its constituent amino acids matched their ground truth counterparts. We denote $N^{\text{pep}}_{\text{match}}$ as the number of peptides with all amino acids correctly matched in a given dataset. Peptide recall was then defined as $N^{\text{pep}}_{\text{match}}/N^{\text{pep}}_{\text{all}}$, where $N^{\text{pep}}_{\text{all}}$ represents the total number of peptides in the dataset. These metrics are essential for quantifying the performance of our predictive algorithms in mass spectrometry data analysis.

**Baselines.** To rigorously assess our model's performance, we conducted a comparative analysis against several state-of-the-art methods. The baselines are categorized by architecture: Peaks (Ma et al., 2003), representing the Database (DB) approach. The Autoregressive (AR) methods in-

| Metrics | Category | Methods | Mouse | Human | Yeast | M. mazei | Honeybee | Tomato | Rice bean | Bacillus | C. bacteria | **Average** |
|---|---|---|---|---|---|---|---|---|---|---|---|---|
| | DB | Peaks Novo (Ma et al., 2003) | 0.600 | 0.639 | 0.748 | 0.673 | 0.633 | 0.728 | 0.644 | 0.719 | 0.586 | 0.663 |
| | | Deep. (Tran et al., 2017) | 0.623 | 0.610 | 0.750 | 0.694 | 0.630 | 0.731 | 0.679 | 0.742 | 0.602 | 0.673 |
| | | Point. (Qiao et al., 2021) | 0.626 | 0.606 | 0.779 | 0.712 | 0.644 | 0.733 | 0.730 | 0.768 | 0.589 | 0.687 |
| AA Precision | AR | Casa. (Yilmaz et al., 2022) | 0.689 | 0.586 | 0.684 | 0.679 | 0.629 | 0.721 | 0.668 | 0.749 | 0.603 | 0.667 |
| | | Ada. (Xia et al., 2024b) | 0.646 | 0.618 | 0.793 | 0.728 | 0.650 | 0.740 | 0.719 | 0.739 | 0.642 | 0.697 |
| | | Casa.V2 (Yilmaz et al., 2024) | 0.760 | 0.676 | 0.752 | 0.755 | 0.706 | 0.785 | 0.748 | 0.790 | 0.681 | 0.739 |
| | NAR | Prime. (Zhang et al., 2025) | 0.784 | 0.729 | 0.802 | 0.801 | 0.763 | 0.815 | 0.822 | 0.846 | 0.734 | 0.788 |
| | | **Ours** | **0.800** | **0.730** | **0.818** | **0.819** | **0.780** | **0.825** | **0.835** | **0.854** | **0.742** | **0.800** |
| | DB | Peaks Novo (Ma et al., 2003) | 0.197 | 0.277 | 0.428 | 0.356 | 0.287 | 0.403 | 0.362 | 0.387 | 0.203 | 0.322 |
| | | Deep. (Tran et al., 2017) | 0.286 | 0.293 | 0.462 | 0.422 | 0.330 | 0.454 | 0.436 | 0.449 | 0.253 | 0.376 |
| | | Point. (Qiao et al., 2021) | 0.355 | 0.351 | 0.534 | 0.478 | 0.396 | 0.513 | 0.511 | 0.518 | 0.298 | 0.439 |
| Peptide Recall | AR | Casa. (Yilmaz et al., 2022) | 0.426 | 0.341 | 0.490 | 0.478 | 0.406 | 0.521 | 0.506 | 0.537 | 0.330 | 0.448 |
| | | Ada. (Xia et al., 2024b) | 0.467 | 0.373 | 0.593 | 0.496 | 0.431 | 0.530 | 0.546 | 0.528 | 0.372 | 0.481 |
| | | Casa.V2 (Yilmaz et al., 2024) | 0.483 | 0.446 | 0.599 | 0.557 | 0.493 | 0.618 | 0.589 | 0.622 | 0.446 | 0.539 |
| | NAR | Prime. (Zhang et al., 2025) | 0.567 | 0.574 | 0.697 | 0.650 | 0.603 | 0.697 | 0.702 | 0.721 | 0.531 | 0.638 |
| | | **Ours** | **0.583** | **0.581** | **0.709** | **0.667** | **0.616** | **0.705** | **0.720** | **0.736** | **0.549** | **0.653** |

*Table 1.* Comparison of the performance on the 9-species-V1 benchmark datasets. The models are categorized by their architecture type: DB represents Database, AR stands for Autoregressive Generation, and NAR denotes Non-Autoregressive Generation. The bold font indicates the best performance.

clude DeepNovo (Tran et al., 2017), which integrates CNN and LSTM architectures; PointNovo (Qiao et al., 2021), which processes mass spectrometry data across varying resolutions without increased computational complexity; Casanovo (Yilmaz et al., 2022), a transformer-based model; CasanovoV2 (Yilmaz et al., 2024), which enhances sequencing accuracy by incorporating beam search; and AdaNovo (Xia et al., 2024b), which leverages Conditional Mutual Information to achieve better AR performance. Lastly, PrimeNovo (Zhang et al., 2025), the first Non-Autoregressive Generation (NAR) method, sets a new benchmark for peptide sequencing precision.

**Model Details.** We starts by transforming all inputs, such as peaks, precursors, peptides, and amino acids, into a 400-dimensional embedding space, which serves as the foundation for further processing within the model. Built upon this embedding space, the model's architecture features a 9-layer Transformer, where each layer contains eight attention heads. The feedforward network across all attention layers has a dimension of 1024. To optimize the model's performance, spectra were processed with a batch size of 1600 during the training phase. An initial learning rate of 4e-4 was implemented, which was gradually increased to the target peak within the first epoch and then followed a cosine decay schedule to ensure a controlled reduction over time. Model parameters were optimized using the AdamW optimizer (Kingma & Ba, 2014). This strategic learning rate adjustment was crucial during the 30 epochs of training, which were conducted using eight A100 GPUs. The model's hyperparameters, such as the number of layers, em-

bedding dimensions, attention heads count, and learning rate scheduling strategy, were consistently applied as default settings for all subsequent downstream experiments unless modifications were necessary. More information on its implementation details can be referred to the code in our provided link.

### 4.2. Results

**Performance on 9-species-V1 Benchmark Dataset.** In this study, we assess the performance of various models on the 9-species-V1 benchmark dataset. The experimental results, summarized in Table 1, indicate that our proposed model demonstrates superior performance in amino acid precision across most tested species, achieving an average precision of 0.800. Furthermore, our model excels in peptide recall, with an average recall rate of 0.653.

Notably, our model surpasses all other methods in both amino acid precision and peptide recall for 8 out of the 9 species tested. In these species, our model achieves the highest amino acid precision, ranging from 0.780 to 0.854, and the highest peptide recall, ranging from 0.549 to 0.736. This performance underscores the robustness and generalizability of our approach across a diverse set of species. For the human species, our NAR model achieves an amino acid precision of 0.730 and a peptide recall of 0.581. Although these results are slightly lower than those of the state-of-the-art AR model ContraNovo, they significantly narrow the gap between NAR and AR models.

Overall, the superior performance of our model demon-

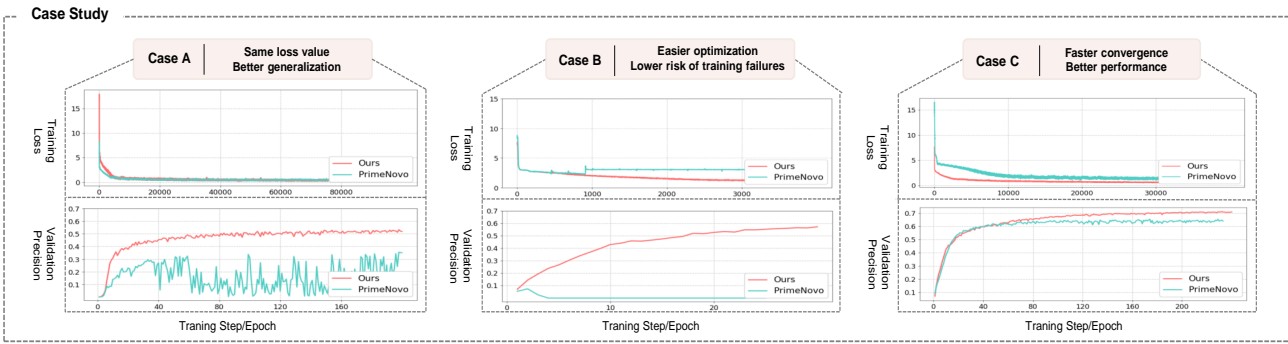

*Figure 4.* A case study showing three types of training failures that frequently happened during the training of NAT peptide sequencing models. RefineNovo can successfully mitigate training problems in traditional NAT models. Note that the plots show training on different datasets rather than one single training for RefineNovo, and validation does not involve looking at true tokens. The training data logging was done and supported by Neptune.AI.

strates its capability to effectively identify peptides in complex mass spectrometry data, thereby highlighting the potential of our approach to advance proteomics research.

**Performance on 9-species-V2 Benchmark Dataset.** We comprehensively evaluate our NAT model's performance against state-of-the-art baselines on the 9-species-V2 dataset, as detailed in Table 2. Our model consistently excels in amino acid precision and peptide recall across most species. Specifically, it achieves the highest amino acid precision for all 9 species, averaging 0.907, surpassing previous NAT and AT models. In peptide recall, our model leads in 8 species, with an average of 0.790, also outperforming NAT baselines. Although ContraNovo marginally outperforms us in human recall, our model demonstrates its generalizability and overall superiority. These results confirm our model's robustness in peptide identification.

**Training Success Rate and Case Study.** Our approach aims to ease the learning difficulty in early-stage training and improve overall learning outcomes. We conducted a comprehensive case study demonstrating how RefineNovo assists with the training process and facilitates generalization. Specifically, we showcase three types of training obstacles frequently encountered during the training of the baseline NAT model, PrimeNovo, and compare the training loss and validation accuracy with ours. We demonstrating training on two different training datasets as adopted by Mao et al. (2023) and Yilmaz et al. (2022). All model configurations were based on the optimal settings and were trained under the same environment, batch size, and learning rate. As shown in 4, PrimeNovo frequently (up to 80% of the time on some datasets) suffers from loss explosion (Case B), heavily depending on the choice of random seed for good convergence. In contrast, our model eliminates the problem of loss explosion due to its easy learning objective at the beginning stage by peeking at true tokens.

When both models ensure smooth loss convergence (Case A), PrimeNovo often suffers from overfitting issues with oscillating validation accuracy, suggesting the model learns to memorize rather than generalize. RefineNovo's gradual training strategy leads to much smoother validation accuracy. Moreover, RefineNovo often exhibits higher learning speed and generalization outcomes (Case C), demonstrating the superiority of the proposed approach in training.

Lastly, we systematically evaluate training stability by randomly selecting 20 different subsets from the MassiveKB dataset and training both PrimeNovo and our model on each split. Among the 20 training runs, PrimeNovo fails in 18 cases due to loss explosion or extreme overfitting, while our model fails only once. This demonstrates that our approach is 90% more likely to result in successful training, establishing a significantly more stable and reliable training paradigm.

**Performance of RefineNovo for Similar Mass Amino Acids.** Distinguishing amino acids with similar molecular weights is crucial for accurate peptide sequencing. Glutamine (Q) and Lysine (K) differ by just 0.036385 Da, while oxidized Methionine (Met(O)) and Phenylalanine (F) have nearly identical weights, challenging algorithmic differentiation. We rigorously evaluated RefineNovo's proficiency in predicting these amino acids, aiming to ascertain its effectiveness in intricate cases. Figure 3 highlights RefineNovo's outstanding performance, consistently achieving superior accuracy in identifying amino acids with minimal mass differences, demonstrating its unparalleled precision.

**Ablation Study.** Table 4 presents an ablation study assessing the impact of three key components on our model's performance. Our baseline is naive NAT model, PrimeNovo, using simple ctc loss. Introducing the curriculum learning with a fixed mask ratio of 0.7 results in a performance decline, indicating that a fixed mask ratio may not be optimal.

| Metrics | Architect | Methods | Mouse | Human | Yeast | M.mazei | Honeybee | Tomato | Rice bean | Bacillus | C.bacteria | **Average** |
|---|---|---|---|---|---|---|---|---|---|---|---|---|
| AA Precision | AT | Casa.V2 (Yilmaz et al., 2024) | 0.813 | 0.872 | 0.915 | 0.877 | 0.823 | 0.891 | 0.891 | 0.888 | 0.791 | 0.862 |
| | NAT | Prime. (Zhang et al., 2025) | 0.839 | 0.893 | 0.932 | 0.908 | 0.862 | 0.909 | 0.931 | 0.921 | 0.827 | 0.891 |
| | | **Ours** | **0.850** | **0.921** | **0.941** | **0.921** | **0.879** | **0.916** | **0.931** | **0.942** | **0.841** | **0.907** |
| Peptide Recall | AT | Casa.V2 (Yilmaz et al., 2024) | 0.555 | 0.712 | 0.837 | 0.754 | 0.669 | 0.783 | 0.772 | 0.793 | 0.558 | 0.714 |
| | NAT | Prime. (Zhang et al., 2025) | 0.627 | 0.795 | 0.884 | 0.812 | 0.742 | 0.824 | 0.837 | 0.849 | 0.626 | 0.777 |
| | | **Ours** | **0.637** | **0.805** | **0.895** | **0.827** | **0.762** | **0.829** | **0.862** | **0.856** | **0.637** | **0.790** |

*Table 2.* Comparison of the performance on the 9-species-V2 benchmark datasets. AT stands for Autoregressive Transformer and NAT stands for non-autoregressive Transformer.

| Methods | Amino Acid Precision | | | |
|---|---|---|---|---|
| | M(O) | Q | F | K |
| Casa.V2 | 0.463 | 0.648 | 0.678 | 0.689 |
| Prime. | 0.578 | 0.770 | 0.806 | 0.800 |
| **Ours** | **0.600** | **0.782** | **0.816** | **0.810** |

*Table 3.* Comparison of precision for amino acids with similar masses.

| Curriculum Learning | Iterative Refinement | Difficulty Annealing | Amino Acid Precision | Peptide Recall |
|---|---|---|---|---|
| | | | 0.788 | 0.638 |
| ✓ | | | 0.733 | 0.558 |
| ✓ | ✓ | | 0.742 | 0.571 |
| ✓ | | ✓ | 0.793 | 0.645 |
| ✓ | ✓ | ✓ | **0.800** | **0.653** |

*Table 4.* Results of the ablation study showing the effects of three key components on RefineNovo's final performance.

Adding iterative refinement with a fixed mask ratio yields a slight improvement, suggesting that iterative refinement can help correct errors. When combining the curriculum learning with difficulty annealing (dynamic masking ratio), we observe a significant enhancement over baseline model. This proves the importance of annealing ratio in adjusting learning curves. When all three components are integrated, our model achieves the highest performance. These results underscore the importance of each component and their synergistic effect, allowing our model to generate highly accurate peptide sequences.

## 5. Conclusion

In conclusion, we introduce RefineNovo, a novel and effective approach to de novo peptide sequencing that sets a new performance benchmark in the field. Our model achieves superior performance and consistently surpasses previous deep learning models across all evaluated metrics.

The model's enhanced training stability and the innovative integration of iterative refinement with the learnt curriculum embedding are key contributors to its effectiveness. These advancements establish RefineNovo as a valuable tool for proteomics research and other downstream tasks.

## Impact Statement

Peptide sequencing plays a fundamental role in proteomics, yet existing computational methods face significant challenges in efficiency, accuracy, and robustness. Our work introduces a novel curriculum learning framework tailored for non-autoregressive Transformers (NATs), addressing the longstanding difficulties of training stability and convergence in CTC-based models. By dynamically adjusting learning difficulty and integrating an iterative refinement strategy, our approach not only enhances model generalization but also significantly reduces training failures, improving both sequence prediction accuracy and reliability.

This research bridges the gap between structured learning paradigms and modern deep learning architectures, paving the way for more efficient peptide sequencing pipelines. Our methodology has the potential to accelerate discoveries in proteomics, drug development, and biomolecular analysis, where high-throughput, accurate sequencing is crucial. By releasing our implementation and trained models, we aim to provide a scalable, reproducible, and widely applicable solution to the broader scientific community.

## Acknowledgement

This project was partially supported by Shanghai Artificial Intelligence Laboratory (S.S.). This work is partially supported by Netmind.AI and ProtagoLabs Inc. This work is also partially supported by CURE (Hui-Chun Chin and Tsung-Dao Lee Chinese Undergraduate Research Endowment) (24924), and National Undergraduate Training Program on Innovation and Entrepteneurship grant(24924).

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

## A. Curriculum Learning

Curriculum learning improves training stability and convergence in CTC-based non-autoregressive (NAT) models by dynamically adjusting the learning difficulty based on model performance. Instead of forcing the model to learn complex sequences from the start, our method progressively increases the difficulty, ensuring a smoother optimization process.

The approach consists of four main stages. First, we compute the best alignment path using the CTC loss function, which determines the optimal token mapping between the predicted and target sequences. Next, we extract an oracle sequence, selecting key reference tokens that serve as easier learning targets for the model. To control learning difficulty, we introduce a dynamic masking ratio, where the probability of masking increases as the model's accuracy improves. Finally, we apply conditional masking, feeding the model partially masked sequences to facilitate a structured learning process that transitions from high supervision to a more generalized sequence prediction task.

The following code provides a Python implementation of our curriculum learning approach, maintaining consistency with Algorithm 1. We use PyTorch to efficiently compute CTC alignments.

```python
if mode == "train":
    with torch.no_grad():
        # Forward pass without gradient updates
        word_ins_out, tgt_tokens, _ = self._forward_step(*batch)
        nonpad_positions = tgt_tokens.ne(self.decoder.get_pad_idx())
        target_lens = nonpad_positions.sum(1)
        pred_tokens = word_ins_out.argmax(-1)
        out_lprobs = F.log_softmax(word_ins_out, dim=-1)

        # Compute sequence lengths
        seq_lens = torch.full(
            (pred_tokens.size(0),), pred_tokens.size(1)
        ).to(self.device)

        # Compute best alignment using CTC
        best_aligns = best_alignment(
            out_lprobs.transpose(0, 1), tgt_tokens, seq_lens,
            target_lens, self.decoder.get_blank_idx(),
            zero_infinity=True
        )

        # Generate oracle sequence
        best_aligns_pad = torch.tensor(
            [a for a in best_aligns], device=word_ins_out.device
        )
        oracle_pos = (best_aligns_pad // 2).clip(
            max=tgt_tokens.shape[1] - 1
        )
        oracle = tgt_tokens.gather(-1, oracle_pos)
        oracle_empty = oracle.masked_fill(
            best_aligns_pad % 2 == 0, self.decoder.get_blank_idx()
        )

        # Compute dynamic masking ratio
        same_num = (pred_tokens == oracle_empty).sum(1)
        keep_prob = ((seq_lens - same_num) / seq_lens * peek_factor)
        keep_prob = keep_prob.unsqueeze(-1)

        # Generate curriculum learning mask
        keep_word_mask = (
```

```
        torch.rand(pred_tokens.shape, device=word_ins_out.device)
        < keep_prob
).bool()
glat_prev = oracle_empty.masked_fill(
    ~keep_word_mask, self.decoder.get_mask_idx()
)
```

## B. Iterative Refinement

### B.1. Pseudo Code

Algorithm 2 outlines our iterative refinement approach. Instead of relying solely on a single-pass decoding, our method iteratively refines the generated peptide sequence by reintroducing previously decoded outputs as conditional input. At each iteration, the decoder takes the spectra and precursor information as input, along with the pseudo-label generated from the previous iteration. The model then refines its predictions by leveraging learned token embeddings, progressively improving sequence accuracy. This iterative process continues for a fixed number of steps, ensuring better alignment between the predicted and true sequences while mitigating common decoding errors.

---

**Algorithm 2** Curriculum-Embedding-based Iterative Refinement

---

**Require:** Spectra **spectra**, Precursors **precursors**, Decoder **decoder**, Encoder **encoder**, Iterations $N$
**Ensure:** Refined output sequence **output_decoded**
 1: Initialize **prev** $\leftarrow$ None
 2: Initialize **output_decoded** $\leftarrow$ [ ]
 3: **for** $i = 1$ to $N$ **do**
 4:    # Forward pass with previously decoded sequence as input
 5:    **output_logits**, _, **output_list** $\leftarrow$ **decoder**(None, **precursors**, *$\ast$**encoder**(**spectra**, **precursors**), **prev**)
 6:    # Decode using argmax and update previous predictions
 7:    **prev** $\leftarrow$ $\arg\max($**output_logits**$, -1)$
 8: **end for**
    RETURN **prev**

---

### B.2. Impact of Iterative Steps

Table 5 illustrates the impact of iterative steps on performance metrics. The first iteration establishes a baseline with a peptide recall of 0.728 and an amino acid (AA) precision of 0.848. By the third iteration, peptide recall improves to 0.736 and AA precision to 0.854, demonstrating that our iterative optimization strategy results in more accurate outcomes. However, beyond the third iteration, both metrics plateau at 0.737 for recall and 0.855 for precision. This plateau suggests that while initial iterations yield significant improvements, further iterations provide minimal additional benefits. Therefore, we select three iterations to balance inference cost with model accuracy.

| Iterative Steps | AA Precision | Peptide Recall |
|:---:|:---:|:---:|
| 1 | 0.848 | 0.728 |
| 2 | 0.853 | 0.731 |
| 3 | 0.854 | 0.736 |
| 4 | 0.855 | 0.737 |
| 5 | 0.855 | 0.737 |
| 10 | 0.855 | 0.737 |

*Table 5.* Effect of different beam sizes on RefineNovo.

## C. CTC Loss Calculation for Protein Sequence

Connectionist Temporal Classification (CTC) loss is a widely used objective function in sequence-to-sequence models, particularly for tasks where the alignment between input and output sequences is unknown. In this section, we provide a detailed explanation of how CTC loss is calculated efficiently using dynamic programming (Zhang et al., 2025; Gu et al., 2017).

### C.1. Problem Definition

The goal of CTC is to compute the total probability of all valid alignment paths $\mathbf{y}$ that reduce to the target sequence $\mathcal{A}$, denoted as $\Gamma(\mathbf{y}) = \mathcal{A}$. Let $\mathcal{I}$ represent the input spectrum and $\mathcal{A} = (a_1, a_2, \ldots, a_n)$ be the target sequence of amino acids. Each alignment path $\mathbf{y} = (y_1, y_2, \ldots, y_T)$ must satisfy the CTC reduction rules: 1. Consecutive identical tokens in $\mathbf{y}$ are merged. 2. Placeholder token $\epsilon$ is removed. 3. Identical tokens adjacent to $\epsilon$ are not merged.

CTC maximizes the total probability of all valid paths, given by:

$$P(\mathcal{A}|\mathcal{I}) = \sum_{\mathbf{y}:\Gamma(\mathbf{y})=\mathcal{A}} P(\mathbf{y}|\mathcal{I}), \tag{6}$$

where $P(\mathbf{y}|\mathcal{I}) = \prod_{t=1}^{T} P(y_t|\mathcal{I})$ under the independence assumption. However, directly enumerating all possible paths is computationally infeasible, as the number of valid paths grows exponentially with the length of $\mathcal{A}$ and $\mathbf{y}$. To address this, we use dynamic programming to efficiently compute $P(\mathcal{A}|\mathcal{I})$.

### C.2. Dynamic Programming Formulation

We define $\alpha(\tau, r)$ as the probability of generating the first $r$ amino acids of the target sequence $\mathcal{A}$ using the first $\tau$ tokens in the alignment path $\mathbf{y}$:

$$\alpha(\tau, r) = P(A_{1:r}|S) = \sum_{\mathbf{y}:\Gamma(\mathbf{y}_{1:\tau})=A_{1:r}} P(\mathbf{y}|S), \tag{7}$$

where $S$ represents the input spectrum $\mathcal{I}$. This recursive relationship allows us to compute $P(\mathcal{A}|\mathcal{I})$ efficiently.

The initialization of $\alpha(\tau, r)$ is as follows:

$$\alpha(\tau, 0) = P(y_1 = \epsilon) \cdot P(y_2 = \epsilon) \cdots P(y_\tau = \epsilon), \qquad \forall 1 \leq \tau \leq T, \tag{8}$$

$$\alpha(1, 1) = P(y_1 = a_1), \tag{9}$$

$$\alpha(1, r) = 0, \qquad \text{for } r > 1. \tag{10}$$

### C.3. Recursive Calculation

To compute $\alpha(\tau, r)$ for $\tau > 1$ and $r \geq 1$, we decompose it based on whether the current token $y_\tau$ matches the target amino acid $a_r$, or if it contributes to a blank or repeated token. Using the law of total probability:

$$\alpha(\tau, r) = \begin{cases} \alpha(\tau - 1, r) \cdot P(y_\tau = a_r), & \text{if } a_r = a_{r-1}, \\ \alpha(\tau - 1, r - 1) \cdot P(y_\tau = a_r), & \text{if } a_r \neq a_{r-1}, \\ \alpha(\tau - 1, r) \cdot P(y_\tau = \epsilon), & \text{otherwise.} \end{cases} \tag{11}$$

This recurrence efficiently aggregates the probabilities of all valid paths reducing to $A_{1:r}$.

### C.4. Loss Function

The final CTC loss is defined as the negative log probability of the target sequence $\mathcal{A}$:

$$\mathcal{L}_{\text{CTC}} = -\log P(\mathcal{A}|\mathcal{I}) = -\log \alpha(T, |\mathcal{A}|). \tag{12}$$

### C.5. Practical Considerations

To ensure numerical stability during training, the log-space version of the dynamic programming equations is often used. Additionally, dynamic programming reduces the computational complexity from exponential to linear in terms of $T$ and $n$. This makes CTC loss feasible for large-scale peptide sequencing tasks.

### C.6. Illustration of CTC Path

An example of the valid paths for a target sequence $\mathcal{A} = (A, T, C)$ and alignment length $T = 5$ is shown below:

- Path 1: $A \ A \ T \ T \ C$

- Path 2: $A \ \epsilon \ T \ \epsilon \ C$

- Path 3: $\epsilon \ A \ T \ C \ \epsilon$

Only paths that reduce to the exact target sequence according to $\Gamma(\cdot)$ are considered valid.

Dynamic programming significantly simplifies the computation of CTC loss by avoiding explicit enumeration of paths. This approach ensures efficiency and scalability in de novo peptide sequencing tasks, enabling the model to focus on meaningful alignment paths while discarding invalid ones. The detailed recursive formulation provided here serves as the foundation for implementing robust CTC-based training for sequence generation models.

## D. Precise Mass Control (PMC) Method

### D.1. Overview

Precise Mass Control (PMC) (Zhang et al., 2025) is a knapsack-like dynamic programming algorithm designed to enforce mass constraints during peptide decoding. The goal of PMC is to ensure that the total mass of the generated peptide sequence aligns with the experimentally measured precursor mass $m_{\text{pr}}$, within a predefined error tolerance $\sigma$. This ensures that the generated peptide is both accurate and physically valid, addressing challenges in non-autoregressive sequence generation where no direct mechanism exists to enforce mass constraints during decoding.

### D.2. Problem Formulation

The PMC problem can be formalized as maximizing the total log probability of generating a peptide sequence $\mathcal{A} = (a_1, a_2, \ldots, a_n)$, subject to a mass constraint:

$$\max_{\mathcal{A}} \sum_{i=1}^{n} \log P(y_i | \mathcal{I}), \tag{13}$$

where $\mathcal{I}$ represents the input spectrum, $P(y_i | \mathcal{I})$ is the model's predicted probability for amino acid $y_i$ at position $i$, and the mass constraint is given by:

$$m_{\text{pr}} - \sigma \leq \sum_{a_i \in \mathcal{A}} u(a_i) \leq m_{\text{pr}} + \sigma, \tag{14}$$

where $u(a_i)$ is the mass of amino acid $a_i$. The goal is to find a sequence $\mathcal{A}$ that maximizes the probability while satisfying the mass constraint.

### D.3. Dynamic Programming Approach

To solve this optimization problem, we employ a dynamic programming (DP) table $d_t^{\ell}$, where $t$ denotes the step index and $\ell$ denotes the mass at that step. The DP table stores the most probable sequence of amino acids that satisfies the mass constraint at each step.

#### D.3.1. INITIALIZATION

For the first decoding step ($t = 1$), we initialize the DP table as follows:

$$d_1^{\ell} = \begin{cases} \epsilon, & \text{if } \ell = 0, \\ y_1, & \text{if } u(y_1) \in [\ell - \sigma, \ell + \sigma], \\ \emptyset, & \text{otherwise.} \end{cases} \tag{15}$$

Here, $\epsilon$ represents the empty sequence, $y_1$ is the first amino acid in the sequence, and $u(y_1)$ denotes its mass.

### D.3.2. RECURSION

At each decoding step $t > 1$, the DP table is updated by considering three cases, depending on the nature of the new token $y_t$: 1. If $y_t = \epsilon$, the mass remains unchanged due to CTC reduction. In this case:

$$d_t^\ell = \bigcup_{\gamma \in d_{t-1}^\ell} \gamma \circ \epsilon, \tag{16}$$

where $\circ$ denotes sequence concatenation.

2. If $y_t$ is identical to the previous token, the sequence remains the same, but the mass remains constant:

$$d_t^\ell = \bigcup_{\gamma \in d_{t-1}^\ell} \gamma \circ y_{t-1}. \tag{17}$$

3. If $y_t$ is a new token, the mass increases, and the potential sequences are updated as:

$$d_t^\ell = \bigcup_{\gamma \in d_{t-1}^{\ell-u(y_t)}} \gamma \circ y_t. \tag{18}$$

### D.3.3. MASS-CONSTRAINED UPDATE

To ensure the DP table does not grow excessively large, we retain only the top $B$ sequences with the highest probabilities at each step:

$$d_t^\ell = \text{top}_B \left( \bigcup_{y_t \in \mathcal{Y}} \sum_{\gamma \in d_{t-1}^{\ell-u(y_t)}} P(\gamma) \right), \tag{19}$$

where $\mathcal{Y}$ represents the set of all amino acid tokens.

### D.4. Final Sequence Selection

After completing all decoding steps, the final sequence is selected as the most probable sequence stored in $d_T^\ell$, where $T$ is the total decoding length:

$$\mathcal{A} = \arg\max_{\gamma \in d_T^\ell} P(\gamma). \tag{20}$$

## E. Performance Comparison on NovoBench

To rigorously benchmark model performance, we evaluated RefineNovo and all baseline models on NovoBench (Zhou et al., 2024), a recently released de novo sequencing benchmark dataset. We adhered to the experimental setup and evaluation protocol outlined in the NovoBench publication, consistently utilizing Saccharomyces cerevisiae (yeast) as the test species for all comparative analyses. We directly evaluated our existing pretrained model, which was trained on MassiveKB, against the NovoBench yeast test sets. This approach, while not leveraging benchmark-specific training, provides a robust assessment of our model's generalization capabilities.

For a direct and equitable comparison with PrimeNovo, the prior state-of-the-art Non-Autoregressive Transformer (NAT)-based model, we utilized its publicly available weights. Both RefineNovo and PrimeNovo were evaluated under identical conditions on the NovoBench test data to ensure comparability.

We notice the relatively low performance on the 7-species dataset when directly testing with all pretrained models such as Casanovo, PrimeNovo, and RefineNovo. Upon further investigation, we found that this dataset was generated using MS equipment with precision levels significantly different from those in the MassiveKB training data. This results in a notable distribution mismatch. Nevertheless, despite this domain shift, the pretrained RefineNovo model still demonstrates clearly better performance compared to other models trained on MassiveKB. This highlights the robustness of our method under distributional variation.

*Table 6.* Performance comparison on the NovoBench benchmark (yeast test species). Scores for models marked with * are quoted from the NovoBench paper or original publications. CV denotes cross-validation results from the original PrimeNovo paper. "–" indicates data not available.

| Model | 9Species (yeast) | 7Species (yeast) | HC-PT |
|---|---|---|---|
| Casanovo * | 0.48 | 0.12 | 0.21 |
| InstaNovo * | 0.53 | – | 0.57 |
| AdaNovo * | 0.50 | 0.17 | 0.21 |
| HelixNovo * | 0.52 | 0.23 | 0.21 |
| SearchNovo * | 0.55 | 0.26 | 0.45 |
| PrimeNovo-CV * | 0.58 | – | – |
| Casanovo-pretrained | 0.60 | 0.05 | – |
| PrimeNovo | 0.70 | 0.09 | 0.85 |
| **RefineNovo (ours)** | **0.71** | **0.09** | **0.88** |

## F. Relationship to Prior Work in Iterative Refinement and Difficulty Annealing

The development of robust generative models for sequences, particularly in complex domains like peptide generation, benefits significantly from strategies that enhance output quality and training stability. In this context, our proposed self-refining module and difficulty annealing strategy build upon established concepts while introducing specific innovations tailored to Non-Autoregressive Transformer (NAT) models and Connectionist Temporal Classification (CTC) based decoding.

### F.1. Iterative Self-Refinement

The principle of iteratively refining generated outputs to improve quality is a powerful paradigm. Our post-training self-refining module, integrated into the main NAT architecture, draws inspiration from multi-pass generation techniques employed in notable protein-related models like ESM-3 and AlphaFold2, which leverage iterative processing for enhanced prediction accuracy.

We acknowledge that the motivation for such refinement—improving generation quality through successive rounds of error correction and adjustment—is shared with several existing lines of research. For instance, masked language modeling approaches, including conditional masked language modeling, and discrete or masked diffusion models also employ iterative refinement (Ghazvininejad et al., 2019; Zheng et al., 2023; Sahoo et al., 2024), often by re-predicting masked or noisy portions of a sequence.

Our method, however, introduces a distinct mechanism specific to NAT models utilizing CTC. Instead of directly refining the generated token sequence, our self-refinement module operates on the **CTC path**. A CTC path represents one of many possible alignment and reduction outcomes that can produce the target label sequence. In our framework, an initial forward pass yields a CTC path. This path is then fed back into the model, allowing for iterative adjustment and refinement of this alignment representation in subsequent passes. The final, refined **CTC path** is then used for the reduction to the ultimate output sequence. This **CTC-path-centric refinement** allows the model to explore and optimize the alignment space more effectively within the NAT framework, differentiating it from methods that directly manipulate the sequence tokens during refinement.

### F.2. Difficulty Annealing in Sequence Generation

Curriculum learning, or "easy-to-hard" training strategies, has demonstrated efficacy in various machine learning tasks, including sequence generation. Prior work has explored difficulty annealing primarily at a coarser granularity, such as at the **task level** (learning simpler tasks before more complex ones) or at the **inter-sequence level** (e.g., training on shorter or structurally simpler sequences before progressing to longer or more complex ones, as explored in some protein generation contexts (Ghazvininejad et al., 2019).

To the best of our knowledge, our proposed difficulty annealing strategy is among the first to implement annealing at a **within-sequence** granularity for NAT models with CTC. Our method defines and modulates difficulty *within each training sequence* by controlling the amount of information exposed from its chosen CTC path during training. Each sequence

effectively starts as an "easier" instance by revealing a greater portion of its CTC path. As training progresses, the visibility of this path information is gradually reduced. This reduction exponentially increases the learning difficulty due to the *combinatorial explosion* in the number of valid CTC paths that could correspond to the same target label sequence.

This fine-grained **within-sequence** and **within-path** difficulty annealing is uniquely enabled by our CTC-sampling mechanism, which is specifically designed for this purpose. This approach plays a crucial role in stabilizing the training dynamics of our NAT model and guiding it towards more robust representations.

