# OpenReview forum: "Curriculum Learning for Biological Sequence Prediction: The Case of De Novo Peptide Sequencing"
_ICML.cc/2025/Conference — ICML 2025 poster_

### Official Review · Reviewer_Qeo9 · 2025-03-13

**Overall Recommendation:** 4

**Summary:**

This paper proposes an improved non-autoregressive peptide sequencing model incorporating a structured protein sequence curriculum learning strategy.

**Claims And Evidence:**

The claims made in the submission are supported by clear and convincing evidence.

**Essential References Not Discussed:**

This paper has no essential references not discussed.

**Experimental Designs Or Analyses:**

The experimental section of this paper needs to incorporate more baseline methods[1][2][3] for comparison.

[1] De novo peptide sequencing with InstaNovo: Accurate, database-free peptide identification for large scale proteomics experiments

[2] Bridging the Gap between Database Search and De Novo Peptide Sequencing with SearchNovo

[3] PowerNovo: de novo peptide sequencing via tandem mass spectrometry using an ensemble of transformer and BERT models

**Methods And Evaluation Criteria:**

This field has an open benchmark dataset, and the method should be tested on it.


[1] NovoBench: Benchmarking Deep Learning-based De Novo Peptide Sequencing Methods in Proteomics

**Other Comments Or Suggestions:**

I have no further comments or suggestions.

**Other Strengths And Weaknesses:**

This paper is well-written, and the proposed method is also novel.

**Questions For Authors:**

1. Using the latest benchmark datasets to test the proposed model.
2.  Add more baseline methods as comparison methods.

**Relation To Broader Scientific Literature:**

This paper introduces the NAT decoding technology to the task of de novo peptide sequencing.

**Theoretical Claims:**

This paper has no theoretical claims.

---

> ### Author Rebuttal · Authors · 2025-04-01
>
> We thank the reviewer for their time and thoughtful comments. We have addressed all of your concerns with point-by-point responses below:
>
> > This field has an open benchmark dataset, and the method should be tested on it.
>
> Thank you for pointing this out. We apologize for missing this benchmark dataset earlier, as it is relatively new. In response, we have conducted additional experiments using the **NovoBench** dataset and evaluated our method, **RefineNovo**, alongside previously reported baselines.
>
> Following the experimental setup from the NovoBench paper, we downloaded the test data (from multiple sources as specified by the paper's data source instruction) and compared our method against the results reported in the benchmark, including those of **PrimeNovo** and other baselines.
>
> Consistent with NovoBench’s evaluation protocol, we used **yeast** as the test species. Due to time constraints during the rebuttal period, we were unable to retrain our model on the NovoBench training data. Instead, we directly evaluated our pretrained model on the benchmark test set.
>
> To highlight the effectiveness of our method, we include a direct comparison between **RefineNovo** and **PrimeNovo**, the previous best-performing NAT-based model. Using the publicly available weights of PrimeNovo from GitHub, we evaluated both models under identical conditions. Our results show that **RefineNovo outperforms PrimeNovo across all test datasets**, as Seen in Table below.
>
> Additionally, to emphasize the performance advantages of NAT-based architectures, we referenced the **cross-validation (CV)** results from the PrimeNovo paper, where the model was trained on nine species (excluding yeast). These results demonstrated the strength of NAT designs relative to other architectures on the same task.
>
> We notice the relatively low performance on the 7-species dataset when directly testing with all pretrained models such as **Casanovo**, **PrimeNovo**, and **RefineNovo**. Upon further investigation, we found that this dataset was generated using MS equipment with precision levels significantly different from those  in the **MassiveKB** training data. This results in a notable distribution mismatch.
> Nevertheless, despite this domain shift, the pretrained **RefineNovo** model still demonstrates clearly better performance compared to other models trained on MassiveKB. This highlights the robustness of our method under distributional variation.
>
> We thank the reviewer again for bringing this benchmark to our attention. ```We will incorporate all experimental results, dataset references, and baseline comparisons into the final version of the manuscript and ensure all relevant works are properly cited.```
>
>
> | **Model**            | **9Species (yeast)** | **7Species (yeast)** | **HC-PT** |
> |----------------------|----------------------|-----------------------|-----------|
> | Casanovo *           | 0.48                 | 0.12                  | 0.21      |
> | InstaNovo *          | 0.53                 | –                     | 0.57      |
> | AdaNovo *            | 0.50                 | 0.17                  | 0.21      |
> | HelixNovo *          | 0.52                 | 0.23                  | 0.21      |
> | SearchNovo *         | 0.55                 | 0.26                  | 0.45      |
> | PrimeNovo-CV *       | 0.58                 | –                     | –         |
> | Casanovo-pretrained | 0.60                 | 0.05                | –         |
> | PrimeNovo    | 0.70                 | 0.09                  | 0.85      |
> | **RefineNovo**       | 0.71            | 0.09              | 0.88  |
>
> \* **marks numbers are quoted from benchmark/original paper**
>
>
> > The experimental section of this paper needs to incorporate more baseline methods [1][2][3] for comparison.
>
> Thank you for the suggestion. We will incorporate the **full set of baseline comparisons using the NovoBench benchmark**, as shown in the table above. This includes **InstaNovo**, **AdaNovo**, **HelixNovo**, **SearchNovo**, and **PrimeNovo**. ```These methods will be discussed and properly referenced in both the <Related Work> and <Experiments> sections of the revised manuscript```.
>
> Regarding **PowerNovo**, we have reviewed the published work and found that the dataset used appears to be inconsistent with the NovoBench benchmark. Nonetheless, we will discuss PowerNovo separately in both the *Related Work* and *Experiments* sections to ensure a comprehensive comparison and fair contextualization of our approach.
>
>
> **We sincerely thank the reviewer again for highlighting these points—your feedback has helped us significantly improve the completeness and clarity of our experimental evaluation, let us know if you have further questions after reading our responses!!**

---

> > ### Comment · Reviewer_Qeo9 · 2025-04-03
> >
> > Thanks to the author's answer, I have updated my score

---

> > > ### Author Response · Authors · 2025-04-03
> > >
> > > We thank the reviewer for the effort and time again!
> > > authors

---

### Official Review · Reviewer_NQdu · 2025-03-14

**Overall Recommendation:** 4

**Summary:**

The work proposes a non-autoregressive transformer (NAT)-based curriculum learning framework to deduce the animo acid sequence from tandem mass spectrometry signals. The input peak signals are encoded with an transformer encoder, which is then used to predict the probability of tokens on each position. The decoding is trained on the sampled decoding paths from the predicted probability with increasingly higher masked ratios. The predictions are iteratively refined during training. Results indicate higher performance across multiple species than the baseline methods.

**Claims And Evidence:**

The authors provide comprehensive results to support their claims.

1. In the results section, ContraNovo is compared to several times, producing slightly higher performance. However, the actual values are not shown in the main text or supplementary.

**Essential References Not Discussed:**

No

**Experimental Designs Or Analyses:**

The experiments and analyses are sound and valid.

**Methods And Evaluation Criteria:**

The methods and evaluation make sense for the application, despite a few points that needs clarification:

1. Is the masking performed on $y'$ or $A$? Also, how is the CTC objective defined with masking?

2. As the path $y$ does not necessarily have the same length as the ground truth sequence, how is the masking performed? More specifically, how are ground truth tokens from $A$ incorporated into $y$?

3. In the earlier stages of the training, the predicted token probabilities are inaccurate and all paths that satisfy $\Gamma(y)=A$ would have low probability, which would lead to biases in the training. Does the proposed method take into account the probability of the sampled path $y'$?

**Other Comments Or Suggestions:**

See previous sections

**Other Strengths And Weaknesses:**

See previous sections

**Questions For Authors:**

I only have some minor questions and concerns. See previous sections.

**Relation To Broader Scientific Literature:**

The proposed framework combines well-established techniques in the field as well as broader machine learning to enhance the effectiveness of the inference and address several technical challenges, such as the optimization of the CTC objective and the iterative refinement.

**Theoretical Claims:**

I checked the definition of the CTC objective and the PMC.

---

> ### Author Rebuttal · Authors · 2025-04-01
>
> We sincerely thank the reviewer for their time and effort in providing thoughtful comments and feedback. Below, we provide a point-by-point response to your questions and concerns.
>
> > Is the masking performed on $y'$ or $A$.  Also, how is the CTC objective defined with masking?
>
>
> The masking is performed on a selected CTC path $y'$, which is one of the possible alignments (CTC paths) derived from the true label sequence $A$.
>
> For example, if the true label sequence $A$ is **ACTC** and the generation length is 5, there can be multiple valid CTC paths $y$, such as:
>
> - `A C T T C`
> - `A C T C C`
> - `A C ε T C`
> - *...and many others*
>
> Our algorithm proceeds as follows:
>
>  **Forward Pass for CTC Path Selection:**
>    We first perform a forward pass to obtain the most likely CTC path $y'$ corresponding to the ground truth $A$. For instance, the model may choose `A C T T C` (all $y$ are ranked by their total probability for selection).
>
>  **Masking:**
>    We then apply masking to this specific path $y'$. For example, it may become:
>    `A <mask> T <mask> C`
> after masking.
>
>
> **Prediction Condition:**
>    This masked sequence is then used as a conditioning input. The model is tasked with predicting the full distribution over all valid CTC paths for $A$—such as `ACTTC`, `ACTCC`, `AC ε TC`, etc.—given the partially masked version of one such path.
>
> The CTC objective **remains unchanged**. The model is still trained to maximize the total probability over ```all``` valid CTC paths corresponding to the true label $A$. Intuitively, the model sees  ```one```  **partially** revealed  CTC path and is asked to infer and generalize over the ```full``` space of valid CTC paths.
>
>
> > As the path $y$ does not necessarily have the same length as the ground truth sequence $A$, how is the masking performed? More specifically, how are ground truth tokens from $A$ incorporated into $y$?
>
> As illustrated in the example above, masking is applied to **one** single sampled CTC path $y'$ derived from the ground truth sequence $A$. Each $A$ can correspond to $\mathcal{O}(b^n)$ possible paths $y$, where $n$ is the generation length in NAT (40 in our case), and $b$ is the vocabulary size.
>
> Regarding your question: while $A$ and $y$ may differ in length, all sampled $y$ from $\mathcal{O}(b^n)$ have the same fixed generation length (e.g., 40). We sample ```one``` such $y'$, apply masking to it, and use the result as a condition for predicting ```all``` valid paths. In this way, some of the tokens from $A$ naturally appear in the unmasked positions of $y$, providing partial supervision.
>
> For example, let the ground truth be $A = \text{ACTC}$, and a few possible CTC paths $y$ could be:
>
> - `A C T T C`
> - `A C T C C`
> - `A C ε T C`
> - *...among others*
>
> We might select $y'$ = ```ACTCC```, and mask it to get:
> `A <mask> T <mask> C`
>
> This masked sequence includes some ground truth tokens from $A$ and serves as a partial observation. The model is then trained, under the CTC loss, to recover the full output distribution over all valid paths $y$ corresponding to $A$. This encourages it to learn both how to complete the sequence and how to generalize across CTC paths (all of length 40).
>
> > In the earlier stages of training, the predicted token probabilities are inaccurate and all paths that satisfy the CTC constraints would have low probability, which could lead to biases in training. Does the proposed method take into account the probability of the sampled path?
>
> Yes, this is a very insightful question—thank you for raising it.
>
> What we observed is that, during the early stages of training, the model tends to assign disproportionately high probabilities to the $\epsilon$ (blank) token. As a result, the selected CTC path $y$ (via greedy decoding) often includes many $\epsilon$ tokens. These $\epsilon$-heavy paths are selected simply because they appear most probable under the model’s initial, untrained predictions.
>
> To address this, we experimented with **top-$k$ sampling** during early training to encourage more diverse path selection. However, in practice, we observed that this made little difference. As training progresses, the model rapidly learns meaningful path patterns, and the issue of degenerate $\epsilon$-dominated paths diminishes.
>
> Eventually, greedy selection (top-1) naturally yields well-formed CTC paths with fewer consecutive $\epsilon$ tokens. We found almost no performance difference between top-$k$ sampling and greedy choice, especially beyond the initial training phase.
>
> Thank you again for highlighting this point!
>
> **We thank the reviewer for the careful and detailed reading—masking in the context of CTC can indeed be tricky. We will include a more detailed explanation with the examples to avoid any confusion.**
>
> **Please don’t hesitate to reach out with further questions!**
>
> **Thank you again for your time!**

---

### Official Review · Reviewer_V8fu · 2025-03-14

**Overall Recommendation:** 4

**Summary:**

This paper proposed a new method, named RefineNovo, that improves non-autoregressive transformers (NATs) for peptide sequencing by introducing a curriculum learning strategy and a self-refinement module (including a "difficulty annealing" strategy). It reduces training failures by over 90% and enhances sequence accuracy. Authors evaluated on 9-species-v1 and 9-species-2 benchmarks that show improved performance over existing methods.

**Claims And Evidence:**

- The idea of introducing a curriculum learning strategy to peptide sequencing is indeed novel.

- The method is properly explained, with clear depiction, discussion of the components, and mathematical derivations in the appendix.

Concerns:
- The self-refining module has been explored for peptide generation to some extent, e.g., with discrete or masked diffusion language modeling, conditional masked language modeling, etc. As far as I know, the difficulty annealing strategy has been explored is some studies for sequence generation task e.g., conditional masked language modeling (Marjan et al. 2019), LM-design (Zaixiang et al,. 2023). I wonder how the proposed method RefineNovo is different from these approaches.

**Essential References Not Discussed:**

As mentioned before, the main contribution is introducing a curriculum learning strategy and a self-refinement module, which is much related to discrete or masked diffusion language modeling (e.g., Subham et al. 2024) and conditional masked language modeling (Marjan et al. 2019) that the authored did not cite or discuss.

**Experimental Designs Or Analyses:**

The experimental design with the two dataset 9-species-v1 and 9-species-v2 as well as the analyses on training failures are clearly discussed.

**Methods And Evaluation Criteria:**

- The proposed method, focusing on three components, is indeed relevant for the problem at hand.

- The authors show results on ed 9-species-v1 (Tran et al., 2017) and 9-species-v2 (Yilmaz et al., 2024) benchmark datasets. The experimental results reported in Tables 1, 2, and 3 support their claim.

- The author also showed ablation of the effect of three different components in Table 4, which is useful to see get a clearer idea how each component and their combinations are affecting the generation task.

- In Figure 4, they also show a case study on training failures, where RefineNovo is clearly showing promising improvements.

**Other Comments Or Suggestions:**

N/A

**Other Strengths And Weaknesses:**

N/A

**Questions For Authors:**

N/A

**Relation To Broader Scientific Literature:**

The main contribution of this paper is related to the research on bio-sequencing (especially for proteins), as well as machine learning research on bio-sequence generation. There are prior works on protein sequence generation, especially in the regime of inverse folding, where ideas closely related to what was proposed here has been explored to some extend.

**Theoretical Claims:**

- The main paper does not have any theoretical claims.

- The appendix has discussion on CTC loss computation and PMC methods, which looks okay. Although there are not the contribution of this paper, I appreciate the authors including that in the appendix for a clearer picture.

- The appendix also include the algorithms for Curriculum Learning and Iterative Refinement, which also are properly written.

Minor concerns:
- In Equation 3 (page 4), there is a wrong math notation: sum over probabilities is written to be directly equal to the sum over log probabilities.

- There is notation mismatch between Equations 3 and 4.  Equation 4 uses L as the likelihood, and Equation 3 used L as the negative log likelihood. I would suggest using different symbols for clarity.

---

> ### Author Rebuttal · Authors · 2025-04-01
>
> We thank the reviewer for the thoughtful review and for recognizing the novelty and effectiveness of our proposed method. We sincerely appreciate your effort and provide below point-by-point responses to your comments.
>
> > The self-refining module has been explored for peptide generation to some extent
>
> We thank the reviewer for their interest in comparing our "difficulty annealing" and "self-refinement" strategies with related prior work ```(Marjan et al., 2019; Zaixiang et al., 2023; Subham et al., 2024)```.
>
> Regarding self-refinement, our method introduces a **post-training refining module** integrated into the main NAT architecture. As noted in our paper, it is **inspired by prior work such as ESM-3 and AlphaFold2**, which leverage multi-pass generation for more accurate predictions in protein-related tasks. We reviewed the works mentioned by the reviewer and **agree that masked modeling and diffusion-based approaches share a similar motivation**—improving generation quality through iterative refinement.
>
> We acknowledge that we did not adequately cite these related works in the current draft and **will revise the manuscript to include and discuss them**. Specifically, our refinement method is designed as a CTC-path refinement, specific to **NAT models**. Unlike prior methods that directly refine the generated sequence, our approach refines the CTC path obtained from a previous forward pass. This CTC path represents _**one**_ possible reduction outcome under CTC alignment, and our refinement allows the model to iteratively adjust it in future passes. The final path will be used for reduction of final sequence. **The core motivation remains the same with many prior work**: to improve final predictions through successive rounds of error correction and adjustment, as mentioned prior work. **We will make this clear in final paper with proper reference to all above mentioned paper and others**!
>
> > Difficulty annealing strategy has been explored is some studies for sequence generation task e.g., conditional masked language modeling
>
> Regarding difficulty annealing, to the best of our knowledge, we are **among the first to apply _within-sequence_ difficulty annealing**. Most prior work using "easy-to-difficult" curriculum strategies does so at a **coarser granularity**—typically at the _task level_, where simpler tasks are learned before more challenging ones. In some protein-related work, including those mentioned by the reviewer, the approach is _intra-sequence_—e.g., learning shorter or simpler sequences before longer, more complex ones.
>
> In contrast, our method defines difficulty **within each sequence** by modulating the **amount of CTC path information exposed during training**. Each sequence can start with an easier version—by revealing more information in its chosen CTC path—and gradually become harder by reducing path visibility. This increases difficulty exponentially due to the combinatorial number of valid CTC paths per label. Our design thus enables _within-sequence_ and _within-path_ difficulty annealing, made possible by our CTC-sampling mechanism tailored for this purpose. This fine-grained annealing plays a key role in our model’s training dynamics. We will further discuss this and difference with previous work ```(Marjan et al., 2019; Zaixiang et al., 2023; Subham et al., 2024)```
>  in our final writing!
>
> > In Equation 3 (page 4), there is a wrong math notation: sum over probabilities is written to be directly equal to the sum over log probabilities.
>
> Thank you for the careful reading! We will ensure this error is corrected in the final version. It should read: the log of the product of all token probabilities equals the sum of the log of each token’s probability. Thanks again for catching this!
>
> > There is notation mismatch between Equations 3 and 4. Equation 4 uses L as the likelihood, and Equation 3 used L as the negative log likelihood. I would suggest using different symbols for clarity.
>
> Apologies for the confusion—we inadvertently reused the same notation without realizing it. We will revise the notation to use distinct symbols to avoid ambiguity. Thank you again for the careful reading!
>
> > a self-refinement module, which is much related to discrete or masked diffusion language modeling (e.g., Subham et al. 2024) and conditional masked language modeling (Marjan et al. 2019) that the authored did not cite or discuss.
>
>  we will thoroughly discuss the similarities and differences between their self-refinement modules  (Marjan et al., 2019; Zaixiang et al., 2023; Subham et al., 2024) and ours based on above discussion, and ensure all mentioned and related works are properly cited in the Related Work section. Thank you for pointing this out!
>
>
> **Please let us know if you have any further questions—we’d be happy to address them. Thanks again for your time and thoughtful review!**

---

> > ### Comment · Reviewer_V8fu · 2025-04-08
> >
> > Thanks to the authors for their clear explanation and addressing the comments. I am keeping the previously assigned (high) score.

---

> > > ### Author Response · Authors · 2025-04-08
> > >
> > > We thank the reviewer for their confirmation and time in reviewing our work!
> > > ~Authors

---

### Decision · Program_Chairs · 2025-05-01

**Decision:**

Accept (poster)

**Comment:**

This paper presents RefineNovo, a non-autoregressive Transformer model (RefineNovo) for de novo peptide sequencing, enhanced with a curriculum learning strategy and a self-refinement module. All three reviewers recommend acceptance, citing the method’s novelty, sound design, and strong experimental results across multiple datasets.

While some related work in curriculum learning and refinement is not deeply discussed, and additional baseline comparisons (e.g., from NovoBench) could further strengthen the study, the reviewers were satisfied by the author's rebuttal and overall contributions. The paper is clearly written, well-motivated, and offers a practical improvement for peptide sequencing.